# HiSpec: Hierarchical Speculative Decoding for LLMs

## Abstract

Speculative decoding accelerates LLM inference by using a smaller draft model to speculate tokens that a larger target model verifies. Verification is often the bottleneck (e.g. verification is $4\times$ slower than token generation when a 3B model speculates for a 70B target model), but most prior works focus only on accelerating drafting. *"Intermediate"* verification reduces verification time by discarding inaccurate draft tokens early, but existing methods incur substantial training overheads in incorporating the intermediate verifier, increase the memory footprint to orchestrate the intermediate verification step, and compromise accuracy by relying on approximate heuristics.

We propose *Hierarchical Speculative Decoding (HiSpec)*, a framework for high-throughput speculative decoding that exploits *early-exit (EE) models* for low-overhead intermediate verification. EE models allow tokens to exit early by skipping layer traversal and are explicitly trained so that hidden states at selected layers can be interpreted, making them uniquely suited for intermediate verification without drastically increasing compute and memory overheads. To improve resource-efficiency even further, we design a methodology that enables HiSpec to re-use key-value caches and hidden states between the draft, intermediate verifier, and target models. To maintain accuracy, HiSpec periodically validates the draft tokens accepted by the intermediate verifier against the target model. Our evaluations using various representative benchmarks and models show that HiSpec improves throughput by $1.28\times$ on average and by up to $2.01\times$ compared to the baseline single-layer speculation without compromising accuracy.

## 1 Introduction

The deployment of Large Language Models (LLMs) presents inherent trade-offs between critical performance metrics, such as throughput, accuracy, and latency. Typically, larger models produce more accurate and coherent outputs but increase latency and reduce throughput. *Speculative decoding* is an efficient inference technique that improves throughput by employing two models: a smaller *draft* model that *speculates* tokens and a larger *target* model that verifies these tokens. Generating tokens using the draft model improves throughput, whereas verification against the target model ensures high accuracy comparable to that of the target model itself.

Token verification takes significantly longer than draft generation because it involves processing more layers of a larger target model. For example, our studies using 3B and 70B Llama models as the draft and target respectively, shows that verification is $4\times$ slower than draft generation for the ShareGPT dataset with real-user conversations. Long verification latencies severely limit the throughput of speculative decoding because subsequent tokens cannot be generated until the previous tokens have been verified due to the inherent sequential nature of token processing. The severity of this *verification wall* scales with the size of the target models. Unfortunately, most prior works accelerate draft token generation, while the verification bottleneck remains largely unaddressed.

SPRINTER Zhong et al. (2025) reduces the verification latency by performing *"intermediate"* verification using an auxiliary model that discards inaccurate tokens early, also allowing the next round of draft generation to begin sooner. However, while this method improves throughput slightly, it incurs significant computational and memory overheads to orchestrate multiple models simultaneously and degrades accuracy because not all intermediate tokens are verified by the target model.

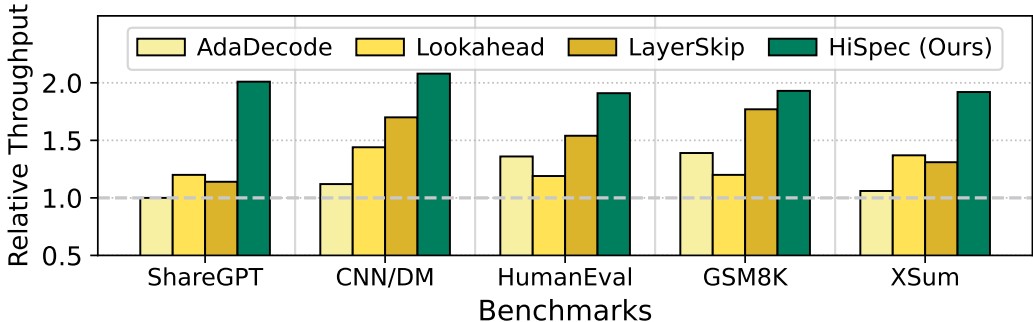

Figure 1: Throughput of various representative benchmarks for the Llama3-8B model relative to standard auto-regressive decoding (*higher throughput is better*). HiSpec consistently outperforms state-of-the-art prior works (AdaDecode, Lookahead Decoding, and LayerSkip) that mainly focus on accelerating draft token generation.

**Our Proposal:** We propose *Hierachical Speculative Decoding (HiSpec)*, a framework to enable low-overhead intermediate verification for high-throughput speculative decoding without sacrificing accuracy. HiSpec uses *early-exit (EE)* models that allow tokens to skip traversing through the entire model by exiting early at selected layers and are explicitly trained so that hidden states at these exit layers can be interpreted. HiSpec exploits this feature to employ early-exit layers for both draft token generation and intermediate verification, thereby overcoming the computational overheads associated with the latter in prior works. To further reduce compute and memory overheads, we propose mechanisms that enable HiSpec to efficiently re-use Key-Value (KV) caches and hidden states across the draft, intermediate verifier, and target layers.

Selecting an appropriate intermediate verifier is critical for overall performance. Using only a few model layers for intermediate verification reduces the token acceptance rate at the target, whereas using a large number of layers reduces throughput. Our studies show that the earlier model layers (up to one-fourth the depth) are critical and generates up to 69% of the response correctly. We use this insight to appropriately position the intermediate verifier that attains a sweet-spot in the throughput versus token acceptance rate trade-off space. Finally, to maintain the same accuracy as the target model and produce outputs consistent with what it generates auto-regressively, HiSpec invokes periodic target (full-model) verification. Our experiments show that HiSpec improves throughput by up to $1.28\times$ on average and by up to $2.01\times$ compared to the baseline single-layer speculation, without compromising accuracy. As shown in Figure 1, HiSpec also consistently outperforms state-of-the-art prior works that improve the performance of speculative decoding by accelerating draft token generation. We also show that HiSpec can be generalized across both pre-trained and post-training modified EE-models, thereby facilitating seamless widespread adoption.

Overall, this paper makes the following contributions.

1. We show that long verification latencies (upto $10.3\times$ higher than draft generation) limit speculative decoding throughput because subsequent tokens cannot be speculated until prior tokens are verified. Most prior works speed up draft generation but do not address the verification wall.

2. We propose *Hierarchical Speculative Decoding (HiSpec)*, a framework that addresses the verification bottleneck by leveraging *early-exit (EE)* models for *low-overhead intermediate verification* so that inaccurate tokens are rejected early. As EE-models are inherently trained to allow tokens to exit at specific model layers, they are uniquely suited for intermediate verification without incurring computational overheads for training the intermediate verifier.

3. We facilitate re-use of key-value caches and hidden states across the draft, intermediate verifier, and target, to avoid redundant computations and improve the resource-efficiency of HiSpec further.

4. To ensure that the output response quality is consistent with what is otherwise produced by the target model auto-regressively, HiSpec also performs periodic target (full-model) verification.

5. Our evaluations show that HiSpec improves throughput by up to $1.28\times$ on average and by up to $2.01\times$ compared to the baseline single-layer speculation, without compromising accuracy.

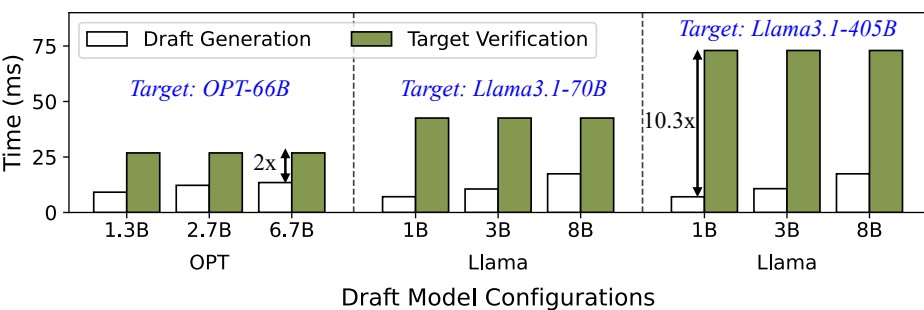

Figure 2: Latency of the draft token generation and token verification phases for different draft and target model combinations for the ShareGPT dataset sha (2023). Verification takes 2-10.3× longer than token generation and the gap between the two latencies grows with the size of the target models.

## 2 BACKGROUND AND MOTIVATION

**Overview of speculative decoding:** Speculative decoding Leviathan et al. (2023); Chen et al. (2023) is an efficient LLM inference technique that splits the inference process into two phases: draft generation and token verification. In the draft generation phase, a smaller and faster but less accurate draft model speculates multiple tokens sequentially based on the preceding context. This is followed by token verification, in which a larger and more accurate target model verifies these predictions by evaluating the likelihood of each draft token against its own distribution. If a draft token does not match the token predicted by the target model, it is discarded along with all subsequent tokens. Otherwise, it is accepted. Note that a round of draft generation begins only after all draft tokens from the previous round have been verified. Speculative decoding enables higher throughput than traditional auto-regressive decoding and ensures that the output is identical to that produced by the target model auto-regressively in isolation.

**Problem:** *Verification wall* **limits speculative decoding performance:** Token verification generally takes significantly longer than draft generation because it involves traversing larger models with complex architectures and more layers. For example, our studies with various Llama and OPT models (ranging between 1B to 405B) show that verification takes 2-10.3× longer than draft generation and accounts for 60-90% of the total response generation time, as shown in Figure 2 (please see Appendix B for more details on our study). We refer to this bottleneck as the *verification wall*. Most importantly, the severity of the verification wall scales with target model sizes and is expected to become even more critical in the future, as we adopt larger models to improve response quality.

**Limitations of prior works:** Several prior works improve speculative decoding performance (discussed in detail in Appendix E on Related Work). However, these approaches mainly optimize token generation by reducing the latency of generating draft tokens and improving the token acceptance rates (percentage of draft tokens accepted by the target model). Consequently, they yield limited returns because they do not reduce verification latencies that dictate overall runtime and throughput.

In a recent work, Zhong et al. propose SPRINTER Zhong et al. (2025) that speeds up token verification via *intermediate verification* using an auxiliary model (which is larger than the draft but smaller than the target model). SPRINTER (1) trains the auxiliary model to mimic the target model using approximate heuristics, (2) employs it to *discard inaccurate tokens early*, and (3) invokes the target model only when absolutely necessary. By saving redundant computations for the early rejected tokens and minimizing the usage of the target model, SPRINTER improves throughput by up to 1.66× for text summarization tasks, emphasizing the benefits of intermediate verification. However, this approach incurs substantial overheads and degrades accuracy. This is because it requires computational resources to train the intermediate verifier. Moreover, orchestrating the draft, intermediate verifier, and the target models simultaneously increases the memory footprint because the weights and key-value (KV) caches of all these models must be stored on the GPU memory. Also, the usage of approximations and infrequent target verification degrades accuracy even for small models.

**Goal- Enable low-overhead intermediate verification without degrading accuracy:** Our goal is to accelerate token verification by enabling *low-overhead and accurate intermediate verification*.

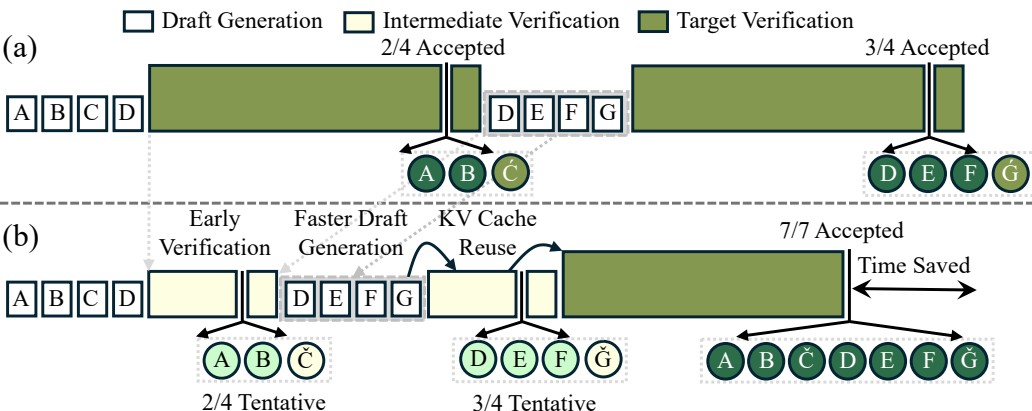

Figure 3: (a) Standard speculative decoding. (b) Our proposal, *HiSpec*, uses *early-exit models* for intermediate verification to reject inaccurate tokens early, thereby also accelerating subsequent draft generation. HiSpec reuses KV caches and hidden states to improve compute and memory efficiency and performs periodic target verification to maintain accuracy.

## 3 DESIGN: HIERARCHICAL SPECULATIVE DECODING (HISPEC)

In this paper, we propose *Hierarchical Speculative Decoding (HiSpec)*, a framework for high-throughput speculative decoding that leverages *early-exit (EE) models* to enable low-overhead intermediate verification. EE-models are a class of LLMs designed to allow tokens to terminate layer traversal and exit at designated exit layer. This is achieved through explicit training to allow hidden states at these selected exit layers to be directly interpreted. HiSpec exploits this feature to use these early exit layers for intermediate verification, thus eliminating any computational and memory overheads associated with integrating an auxiliary intermediate verifier. HiSpec also employs an exit layer (lower than the intermediate verifier) for draft token generation. Thus, the draft and the intermediate verifier corresponds to early exit layers of the EE-model and the final layer (or full model) corresponds to the target. Figure 3 gives an overview of HiSpec. Here, the first round of intermediate verification *tentatively accepts* tokens $A$ and $B$, and rejects token $C$ early. This early verification also naturally enables faster draft generation in the next round, producing tokens $D$, $E$, $F$, and $G$, earlier than in the case of traditional speculative decoding, as shown in the Figure. To produce outputs consistent with what the target model produces auto-regressively, HiSpec also periodically verifies against the full model. Next, we discuss the implementation of HiSpec.

### 3.1 DYNAMIC KV AND HIDDEN STATES MANAGEMENT FOR INTERMEDIATE VERIFICATION

Employing early exits for intermediate verification eliminates any additional training overheads. However, it alone is insufficient to reduce computational overheads because the forward pass for the generation of draft tokens and both intermediate and full-model verification still incurs redundant computations on the same input. To address this issue, HiSpec builds mechanisms to re-use the Key-Value (KV) caches and hidden states across the draft, intermediate verifier, and the target (full model). This is non-trivial because it requires careful alignment of these data structures. Even the slightest misalignment leads to a cascade of errors, which propagates through subsequent layers and token generation steps, ultimately corrupting the generated output.

To overcome this challenge, we take a two-step approach. *First*, during token generation, the generated KV caches and the hidden states are buffered separately until intermediate verification. *Next*, we discard the KV pairs and hidden states associated with the rejected tokens to ensure that the subsequent phase begins with the correct context. HiSpec manages the KV caches and hidden states by expanding them during the draft generation phase and pruning them back after intermediate verification to eliminate entries associated with tokens deemed inaccurate. This approach differs fundamentally from prior single-layer based speculation methods, where these structures are not repeatedly buffered or managed across arbitrary intermediate layers. During target verification, the accumulated hidden states are consumed to compute the acceptance outcome at the final layer, followed by elimination of inaccurate tokens from the KV cache before the next round of draft generation.

## 3.2 POSITIONING INTERMEDIATE VERIFIER LAYER

The positioning of the intermediate verifier layer is critical to maximize the performance of HiSpec. For example, intuitively, using an intermediate verifier layer much higher than the draft layer (*closer to the target*) would improve token acceptance rates, as deeper layers are more accurate and propagate only high-accuracy tokens to the final layer. However, this substantially increases the latency of intermediate verification and reduces throughput. In contrast, selecting an intermediate verifier at a much lower layer (*closer to the draft*), yields high throughput benefits but degrades token acceptance rates because it cannot adequately discard inaccurate tokens.

To position the intermediate layer appropriately, we conduct studies using various models and observe that the early layers are critical and about one-fourth of the model layers generate up to 69% of the response correctly, as shown in Figure 4. Building on this observation, the default implementation of HiSpec positions the intermediate verifier near one-fourth of the model depth, while allocating about one-eight of the layers for draft token generation (half of the intermediate verifier).

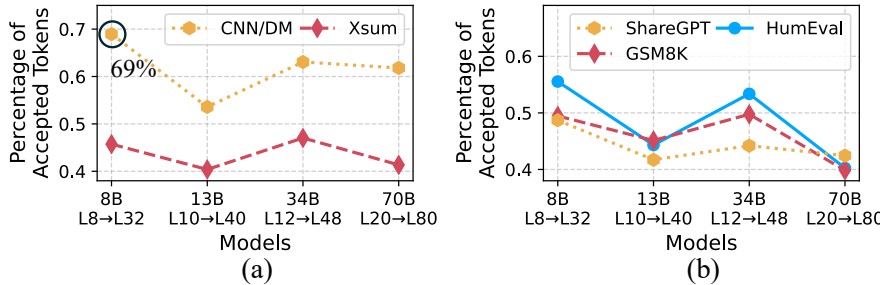

Figure 4: Percentage of tokens produced by one-fourth the model that are accepted by the final layer. We use llama models for (a) text summarization (CNN/DM, Xsum) and (b) other tasks, such as dialogue (ShareGPT), mathematical reasoning (GSM8K), and code generation (HumanEval). Each label denotes one-fourth the model and its final layer (such as $L8 \rightarrow L32$ for Llama-8B with 32 layers). We observe that about one-fourth of the model is sufficient to generate up to 69% of the output tokens correctly. We use this information to position the intermediate verifier in HiSpec.

We conduct additional experiments to verify the efficacy of our selection. We use different exit layers for the draft generation and intermediate verification steps for two different Llama models (with 8B and 70B parameters). Figure 5 shows the HiSpec throughput for each draft and intermediate verifier combination relative to vanilla auto-regressive decoding. We observe that the default configuration offers the highest throughput because it achieves a sweet-spot in the throughput versus token acceptance rates trade-off space. We present additional details about this study in Appendix C.

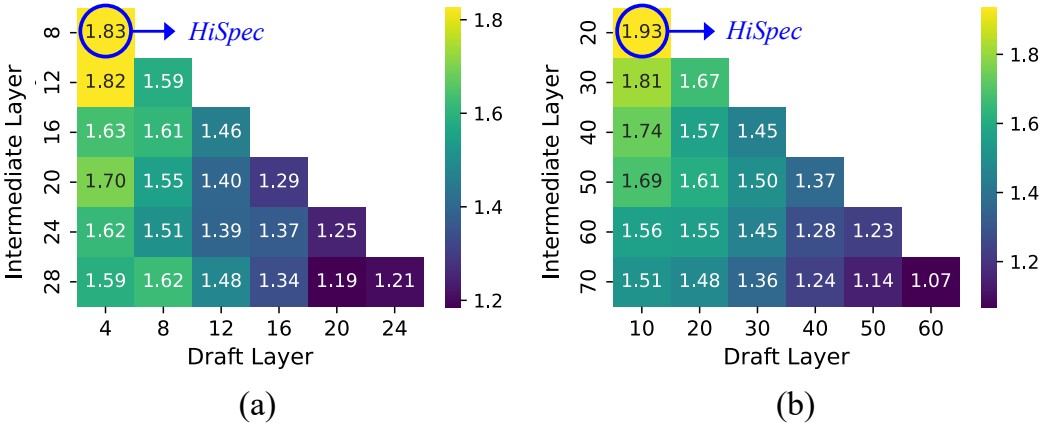

Figure 5: Throughput of HiSpec for different draft and intermediate layer combinations relative to vanilla auto-regressive decoding for the (a) ShareGPT dataset using Llama3-8B (32 layers) and the (b) CNN/DM dataset using Llama2-70B (80 layers). HiSpec's selection of draft and intermediate verifier (circled) yields the highest throughput.

## 3.3 BALANCING INTERMEDIATE AND FULL-MODEL VERIFICATION

Despite the advantages of intermediate verification, target verification is crucial to produce outputs consistent with vanilla auto-regressive decoding. This is because the accuracy of the intermediate verifier is still lower than the target model and it is possible that a token *tentatively accepted* by the intermediate verifier would be eventually rejected by the target. Target verification ensures that such tokens are discarded and the correct ones are regenerated. However, frequent target verification is compute-intensive, slow, and reduces throughput, whereas doing it too infrequently introduces high penalty even if one of the tokens in the tentatively accepted sequence is incorrect, because the entire sequence of tokens following this incorrect token (including itself) is flushed. The probability of this increases with the number of tentatively accepted tokens that are not yet verified by the target.

To retain the accuracy benefits without lowering throughput, HiSpec performs multiple rounds of intermediate verification before invoking the target verification. However, naively invoking target verification every few rounds is sub-optimal because the number of tokens tentatively accepted by the intermediate verifier can vary. Instead, HiSpec employs a dynamic policy that accumulates a sufficient number of tentatively accepted tokens before proceeding with target verification. By default, HiSpec waits until at least four tokens have been tentatively accepted (please see Section 6 for more details). Nonetheless, the intermediate verifier leads to higher token acceptance rates than baseline single-layer speculation methods at the same draft layer. For example, our studies with the Llama2-70B model on the ShareGPT benchmark shows that the average token acceptance rate is only 39.7% when using up to Layer-10 as the draft. HiSpec increases the average token acceptance rate to 58.1% (46% improvement) by employing an intermediate verifier that uses up to Layer-20.

Figure-6 shows how HiSpec efficiently navigates the token acceptance rates and throughput. We consider the CodeLlama-34B model which comprises 48 layers. For the baseline, each datapoint corresponds to a draft with increasing number of layers and we consider all possible choices. Thus, we have 47 draft layer ($L_d$) choices, with a configuration using up to Layer-1, Layer-2, and so on up to Layer-47. As expected, relying on a higher layer for draft generation improves token acceptance rates. However, beyond a certain point ($L_d = 7$), the overall throughput decreases because draft token generation takes too long. Figure-6 also shows the performance for different HiSpec configurations, where each datapoint represents a draft and intermediate verifier combination. As using exit Layer-7 for the draft yields optimal performance in the baseline, we use the same in HiSpec. This leads to 40 possible combinations because now the intermediate verifier can use exit Layer-8, Layer-9, and so on up to Layer-47. We observe that the HiSpec configuration using exit Layer-7 and Layer-12 for the draft and intermediate verifier respectively improves throughput by $1.25\times$ and token acceptance rates by $1.15\times$. This is also consistent with the default HiSpec setting, which uses about one-fourth of the model for intermediate verification and one-eighth for draft token generation.

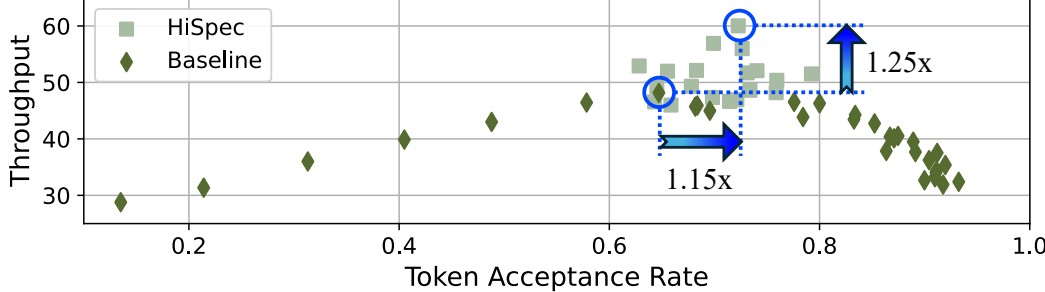

Figure 6: Comparison of token acceptance rates and throughput (*higher is better for both*) for baseline single-layer early-exit speculation (Layerskip) against HiSpec for CodeLlama-34B model (with 48 layers). Each datapoint for the baseline Elhoushi et al. (2024) corresponds to increasing number of layers used for draft token generation (47 possibilities). Each datapoint for HiSpec corresponds to the draft using exit Layer-7 (baseline optimal) and using increasing number of layers for intermediate verification (40 possibilities). HiSpec improves both throughput and token acceptance rates.

Algorithm 1 gives an overview of the HiSpec algorithm and an elaborate version is in Appendix A).

---

**Algorithm 1** Hierarchical Speculative Decoding (HiSpec)

---

1: **Input:** Draft layer $L_d$,
2:     Intermediate layer $L_i$,
3:     Full layer $L_f$,
4:     Context tokens $\mathbf{y}$,
5:     Draft proposal length $N_d$,                                    // Tokens proposed by $L_d$ per step
6:     Tentative acceptance window $N_i$,          // Tokens verified by $L_i$, before verification by $L_f$
7:
8: **while** not end-of-sequence **do**
9:     $\mathbf{V} \leftarrow [\,]$                                   // $\mathbf{V}$ = buffer of tokens verified/accepted by $L_i$
10:     **while** $|\mathbf{V}| < N_i$ **and** not end-of-sequence **do**
11:         $\mathbf{S} \leftarrow \text{GENERATE}(L_d,\ \mathbf{y}\|\mathbf{V},\ N_d)$
12:         $\widehat{\mathbf{V}} \leftarrow \text{LEADINGSUBSTRINGVERIFY}(\mathbf{S},\ L_i,\ [\mathbf{y}, \mathbf{V}])$
13:         $\mathbf{V} \leftarrow [\mathbf{V}, \widehat{\mathbf{V}}]$
14:     **end while**
15:     $\mathbf{F} \leftarrow \text{LEADINGSUBSTRINGVERIFY}(\mathbf{V},\ L_f,\ \mathbf{y})$        // $\mathbf{F}$ = tokens from $\mathbf{V}$ accepted by $L_f$
16:     $\mathbf{y} \leftarrow [\mathbf{y}, \mathbf{F}]$
17: **end while**
18: **return** $\mathbf{y}$
19:
20: **function** LEADINGSUBSTRINGVERIFY(Draft Tokens S, Verifier Layer $L$, Context)
21:     $\widehat{\mathbf{S}} \leftarrow$ longest prefix of $\mathbf{S}$ such that every token $\in$ TOPPREDICTIONS($L$)
22:     **return** $\widehat{\mathbf{S}}$ along with one additional token from $L$
23: **end function**

---

## 4 EVALUATION METHODOLOGY

**Benchmarks:** We evaluate HiSpec for a diverse set of tasks, including real-user conversations (ShareGPT sha (2023)), text summarization (CNN/DM Nallapati et al. (2016) and XSum Narayan et al. (2018)), code generation (HumanEval Chen et al. (2021)) and mathematical reasoning (GSM8K Cobbe et al. (2021)). This is consistent with prior works (more details in Appendix D).

**Setup:** We conduct our experiments on a compute node with four NVIDIA H100 GPUs (94GB HBM3) and an AMD EPYC 9354 CPU, all interconnected via PCIe. We implement HiSpec using the HuggingFace transformers library Wolf et al. (2020), mirroring configuration of various prior works on speculative decoding Elhoushi et al. (2024); Wei et al. (2025); Cai et al. (2024).

**Models:** We evaluate HiSpec across six state-of-the-art models across the Llama2 Touvron et al. (2023), Llama3 Dubey et al. (2024), and the CodeLlama Roziere et al. (2023) families. We use pre-trained and post-training modified *early-exit* variants of these models that are made available by prior works Elhoushi et al. (2024); Wei et al. (2025) on Huggingface hug (2016).

**Baseline:** Prior works primarily accelerate draft generation phase, making them a natural baseline for comparison with HiSpec which focuses on reducing verification overheads. The only prior work on accelerating token verification, SPRINTER Zhong et al. (2025), necessitates training auxiliary verifier models to approximate the target model. Due to the unavailability of these verifier models, excessive computational overheads, and accuracy degradation, we exclude it from our evaluations. We present a more comprehensive discussion of prior work in Appendix E.

▷ **AdaDecode** is the state-of-the-art in early-exit based speculation. It dynamically selects a draft layer at runtime by evaluating the confidence of the intermediate LM heads. However, it necessitates tuning confidence thresholds across benchmarks to achieve robust performance.

▷ **Layerskip** allows more aggressive early exits by statically selecting a single exit-layer to generate draft tokens, making it a strong baseline for benchmarks dominated by simple or predictable tokens.

▷ **Lookahead decoding** modifies the attention mask, enabling the target model to generate multiple draft tokens in parallel. Unlike EE methods, it requires additional FLOPs per decode step.

▷ **SWIFT** takes a different approach by selectively turning-off model layers during draft generation.

## 5  RESULTS

**HiSpec improves throughput:** Table 1 compares the performance of HiSpec against prior works which accelerate draft token generation. HiSpec improves throughput by $1.7\times$ on average and up to $2.08\times$ compared to vanilla auto-regressive decoding. HiSpec consistently outperforms prior works that speeds up draft generation, emphasizing the criticality and benefits of accelerating verification.

Table 1: Throughput of HiSpec and other competing approaches relative to vanilla auto-regressive decoding (*higher is better*), across dialogue (ShareGPT), text summarization (CNN/DM, XSum), code generation (HumanEval), and mathematical reasoning (GSM8K) tasks. These methods do not rely on an auxiliary draft model and ensures that the final outputs are consistent with vanilla auto-regressive decoding. The best performance in each case is highlighted in **bold**.

| Method | Speedup (vs. Vanilla) | | | | |
|---|---|---|---|---|---|
| | ShareGPT | CNN/DM | HumanEval | GSM8K | XSum |
| Llama3-8B$_{\text{INST}}$ | | | | | |
| AdaDecode | $1.00\times$ | $1.12\times$ | $1.36\times$ | $1.39\times$ | $1.06\times$ |
| LayerSkip | $1.14\times$ | $1.70\times$ | $1.54\times$ | $1.77\times$ | $1.31\times$ |
| LookAhead | $1.20\times$ | $1.44\times$ | $1.19\times$ | $1.20\times$ | $1.37\times$ |
| HiSpec *(Ours)* | $\mathbf{2.01\times}$ | $\mathbf{2.08\times}$ | $\mathbf{1.91\times}$ | $\mathbf{1.93\times}$ | $\mathbf{1.84\times}$ |
| Llama2-7B$_{\text{INST}}$ | | | | | |
| LayerSkip | $1.31\times$ | $1.93\times$ | $1.71\times$ | $1.69\times$ | $1.43\times$ |
| LookAhead | $1.37\times$ | $1.34\times$ | $1.24\times$ | $1.50\times$ | $1.43\times$ |
| HiSpec *(Ours)* | $\mathbf{1.70\times}$ | $\mathbf{1.95\times}$ | $\mathbf{1.84\times}$ | $\mathbf{1.92\times}$ | $\mathbf{1.50\times}$ |
| Llama2-13B$_{\text{INST}}$ | | | | | |
| LayerSkip | $1.21\times$ | $1.43\times$ | $1.51\times$ | $\mathbf{1.76\times}$ | $1.29\times$ |
| LookAhead | $1.26\times$ | $1.55\times$ | $1.23\times$ | $1.43\times$ | $1.45\times$ |
| SWIFT | $1.02\times$ | $1.03\times$ | $1.05\times$ | $1.01\times$ | $0.97\times$ |
| HiSpec *(Ours)* | $\mathbf{1.62\times}$ | $\mathbf{1.57\times}$ | $\mathbf{1.52\times}$ | $1.65\times$ | $\mathbf{1.55\times}$ |
| CodeLlama-7B$_{\text{INST}}$ | | | | | |
| LayerSkip | $1.31\times$ | $1.07\times$ | $1.50\times$ | $1.75\times$ | $1.34\times$ |
| LookAhead | $1.35\times$ | $1.58\times$ | $1.23\times$ | $1.36\times$ | $1.46\times$ |
| HiSpec *(Ours)* | $\mathbf{1.49\times}$ | $\mathbf{1.63\times}$ | $\mathbf{1.60\times}$ | $\mathbf{1.75\times}$ | $\mathbf{1.61\times}$ |
| CodeLlama-34B$_{\text{INST}}$ | | | | | |
| AdaDecode | $1.19\times$ | $1.53\times$ | $1.67\times$ | $1.56\times$ | $1.37\times$ |
| LayerSkip | $1.23\times$ | $1.42\times$ | $1.50\times$ | $1.52\times$ | $1.42\times$ |
| LookAhead | $1.31\times$ | $1.50\times$ | $1.18\times$ | $1.32\times$ | $\mathbf{1.49\times}$ |
| SWIFT | $1.09\times$ | $1.13\times$ | $1.24\times$ | $1.12\times$ | $1.05\times$ |
| HiSpec *(Ours)* | $\mathbf{1.55\times}$ | $\mathbf{1.59\times}$ | $\mathbf{1.67\times}$ | $\mathbf{1.60\times}$ | $1.45\times$ |
| Llama2-70B$_{\text{INST}}$ | | | | | |
| LayerSkip | $1.30\times$ | $1.47\times$ | $1.32\times$ | $1.48\times$ | $1.29\times$ |
| SWIFT | $1.10\times$ | $1.17\times$ | $1.12\times$ | $1.09\times$ | $1.10\times$ |
| HiSpec *(Ours)* | $\mathbf{1.64\times}$ | $\mathbf{1.93\times}$ | $\mathbf{1.69\times}$ | $\mathbf{1.72\times}$ | $\mathbf{1.63\times}$ |

**HiSpec is applicable across EE models:** HiSpec heavily exploits early-layers in EE models to improve throughput and can be integrated seamlessly with any EE model with at least two exit layers. Pre-trained EE models usually provide more exit layers, giving more flexibility to HiSpec to optimally position the draft and intermediate verifier for maximal efficiency. For example, Llama checkpoints hosted by Layerskip Elhoushi et al. (2024) which are also used in our evaluations permit tokens to exit at any arbitrary layer. In contrast, post-training EE models contain fewer exit layers, inherently constraining HiSpec's design space and limiting the potential configurations of the draft and intermediate verifier. Nevertheless, HiSpec can still be applied to such models as well and can leverage intermediate exit layers to preemptively reject inaccurate tokens.

# 6 ABLATION STUDY

HiSpec comprises three key design parameters, namely, the number of tokens produced by the draft layer per step ($N_d$), the number of tokens tentatively accepted by the intermediate verifier ($N_i$), and the intermediate verifier layer itself ($L_i$). We analyze the impact of these parameters on HiSpec's throughput using the ShareGPT dataset and Llama3-8B model. As discussed in Section 3.3 (and more details in Appendix C), we use one-fourth of the model as the intermediate verifier because it maximizes both throughput and token acceptance rates (consistent with the default HiSpec). We use this implementation to study the impact of the other two design parameters ($N_d$ and $N_i$).

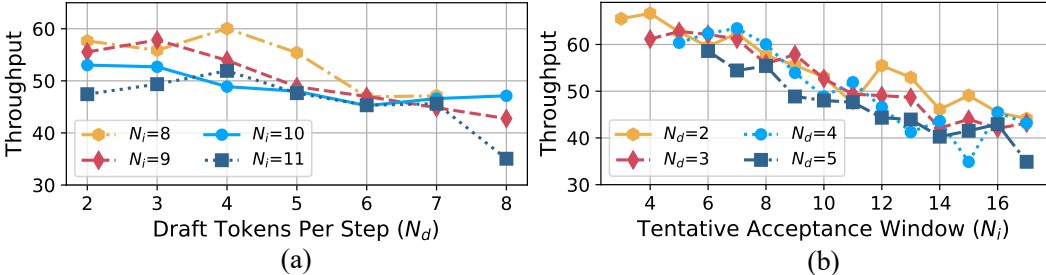

Figure 7: Throughput (*higher is better*) with increasing (a) number of draft tokens per step ($N_d$) and (b) number of tokens tentatively accepted ($N_i$). Lower values of $N_d$ and $N_i$ yield higher throughput, which is expected, because they limit the formation of longer chains of unverified inaccurate tokens.

▷ **Impact of number of draft tokens per step** ($N_d$): A large number of draft tokens per step generally increases the number of tokens flushed in case an incorrect token is identified in the sequence. Moreover, it also increases the latency of intermediate verification as a larger batch of draft tokens must be verified in parallel. Figure 7(a) shows that the throughput decreases with increasing $N_d$, which is consistent with our expectation. The default implementation of HiSpec uses $N_d = 2$.

▷ **Impact of number of tokens tentatively accepted** ($N_i$): Like $N_d$, a large number of tentatively accepted tokens awaiting target verification also increases the number of tokens flushed and regenerated in the event of a verification mismatch. Figure 7(b) shows that the throughput decreases with increasing $N_i$, which is consistent with our expectation. By default, HiSpec uses $N_i = 4$.

# 7 CONCLUSION

Speculative decoding is an efficient LLM inference technique that employs a small draft model to speculate tokens with low latency and a larger target model to verifies these draft tokens with high accuracy. However, its throughput is severely bottle-necked by the token verification step because it takes significantly longer to verify tokens than generate them. Prior works mainly prioritize faster and accurate draft token generation but fail to address the limitations of slow verification.

In this paper, we propose *Hierarchical Speculative Decoding (HiSpec)*, a framework that reduces verification latencies by employing *intermediate verification*. The intermediate verifier tentatively accepts draft tokens that it deems correct, and rejects them otherwise. The early rejection of inaccurate tokens saves computations, whereas the tentatively accepted token sequence serves as the current context for accelerating the subsequent round of draft token generation. HiSpec employs *early-exit (EE) models* that are explicitly trained so that the hidden states at selected model layers can be interpreted, allowing tokens to skip layer traversal and exit earlier than the final model layer. HiSpec exploits this feature of EE-models for low-overhead intermediate verification because it eliminates training overheads in integrating the intermediate verifier. To further improve resource-efficiency, HiSpec proposes mechanisms to reuse key-value caches and hidden states between draft token generation, intermediate verification, and target verification. HiSpec also invokes periodic target verification to ensure that the output sequence produced is consistent with what the target model produces auto-regressively. Our evaluations show that HiSpec improves throughput by $1.28\times$ on average and up to $2.01\times$ compared to baseline single-layer speculation methods, consistently outperforming prior works that mainly accelerate token generation. HiSpec is generalizable across both pre-trained and post-training modified EE-models, facilitating seamless widespread adoption.

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

## A    HISPEC ALGORITHM

Algorithm A describes an elaborate version of the HiSpec design.

---

**Algorithm 2** Speculative Decoding with HiSpec

---

1: **Input:** Draft layer $L_d$,
2:     Full layer $L_f$,
3:     Intermediate layer $L_i \mid L_d < L_i < L_f$
4:     Prompt $= [T_0, T_1, \ldots, T_k]$,
5:     Draft proposal length $N_d$,                          // Tokens proposed by $L_d$ per step
6:     Tentative acceptance window $N_i$,          // Tokens verified by $L_i$, before verification by $L_f$
7:
8: **Algorithm:**
9: **current_context** $\leftarrow$ **prompt**                 // Initializing the context with input tokens
10: $\mathbf{B}_i \leftarrow [\,]$                          // Buffer for tokens accepted by $L_i$ but not committed
11: **while** not end-of-sequence **do**
12:     $\mathbf{ctx}^\star \leftarrow [\mathbf{current\_context}, \mathbf{B}_i]$        // Concatenating context for draft generation cycle
13:     $(\mathbf{speculative\_tokens}, KV_d) \leftarrow L_d.\text{GENERATENEXT}(\mathbf{ctx}^\star, N_d)$
14:     *mismatch* $\leftarrow$ **false**
15:     **for** each lead token $T_j$ in **speculative_tokens do**
16:         $(\mathbf{p}, KV_i) \leftarrow L_i.\text{PREDICTNEXT}(\mathbf{ctx}^\star, KV_d)$
17:         **if** $T_j \in \text{TOPPREDICTIONS}(\mathbf{p})$ **then**
18:             $\mathbf{B}_i \leftarrow [\mathbf{B}_i, T_j]$                              // $L_i$-accepted tokens stay buffered
19:             $\mathbf{ctx}^\star \leftarrow [\mathbf{ctx}^\star, T_j]$
20:         **else**
21:             *mismatch* $\leftarrow$ **true**
22:             $T^\diamond \leftarrow L_i.\text{GENERATENEXT}(\mathbf{ctx}^\star, 1)$       // $1^{st}$ mismatch $\rightarrow$ emit one token from $L_i$
23:             $\mathbf{B}_i \leftarrow [\mathbf{B}_i, T^\diamond]$
24:             **break**                                       // Stop at First Mismatch
25:         **end if**
26:     **end for**
27:
28:     **if** $B_i < N_i$ **then**
29:         **continue**                      // Buffered tokens, $B_i < N_i$, trigger draft generation round
30:     **end if**
31:
32:     **while** $|\mathbf{B}_i| > 0$ **do**                              // Full-model Verification is invoked
33:         $U \leftarrow \mathbf{B}_i[1]$
34:         $\mathbf{q} \leftarrow L_f.\text{PREDICTNEXT}(\mathbf{current\_context}, KV_i)$
35:         **if** $U \in \text{TOPPREDICTIONS}(\mathbf{q})$ **then**
36:             $\mathbf{current\_context} \leftarrow [\mathbf{current\_context}, U]$
37:             $\mathbf{B}_i \leftarrow \mathbf{B}_i[2:]$
38:         **else**
39:             $T^f \leftarrow L_f.\text{GENERATENEXT}(\mathbf{current\_context}, 1)$       // Emit one token from $L_f$
40:             $\mathbf{current\_context} \leftarrow [\mathbf{current\_context}, T^f]$
41:             $\mathbf{B}_i \leftarrow [\,]$           // Flush remaining buffered tokens after full-model verification
42:             **break**
43:         **end if**
44:     **end while**
45: **end while**
46: **return current_context**
47:

---

## B  EVALUATING THE CRITICALITY OF ACCELERATING VERIFICATION

Prior works improves the performance of speculative decoding by accelerating draft generation and verification. We investigate the criticality of accelerating verification comparing its latency against token generation latency. To measure latency, we use the vLLM Kwon et al. (2023) inference engine and various combinations of draft and target models that are selected from prior work Yan et al. (2024); Lasby et al. (2025); Galim et al. (2025). The draft model generates six tokens per iteration. We conducts all these experiments on a node with eight NVIDIA H100 GPUs interconnected by NVlink, evaluated on the ShareGPT dataset. Our evaluations show that accelerating verification is even more critical because it is $2 - 10.3\times$ slower than token generation, as shown in Table 2. Also, the gap between verification and draft generation latencies scale with the size of the target models.

Table 2: Comparison between the latency of draft generation ($T_{\text{gen}}$) and the latency of target verification ($T_{\text{verif}}$)) in speculative decoding. These results show that verification time dominates the total execution time and is significantly slower. Moreover, this gap increases with increasing model sizes..

| Target Model | Draft Model | Latency (ms) | | $T_{\text{verif}}/T_{\text{gen}}$ |
| --- | --- | --- | --- | --- |
| | | Generation ($T_{\text{gen}}$) | Verification ($T_{\text{verif}}$) | |
| facebook/opt-66b | facebook/opt-1.3b | 9.14 | | $2.9\times$ |
| | facebook/opt-2.7b | 12.23 | 26.83 | $2.2\times$ |
| | facebook/opt-6.7b | 13.48 | | $2\times$ |
| Llama-3.1-70B$_{\text{INST}}$ | Llama-3.2-1B$_{\text{INST}}$ | 7.10 | | $6\times$ |
| | Llama-3.2-3B$_{\text{INST}}$ | 10.58 | 42.54 | $4\times$ |
| | Llama-3.1-8B$_{\text{INST}}$ | 17.42 | | $2.4\times$ |
| Llama-3.1-405B$_{\text{INST}}$ | Llama-3.2-1B$_{\text{INST}}$ | 7.10 | | $10.3\times$ |
| | Llama-3.2-3B$_{\text{INST}}$ | 10.58 | 72.97 | $6.9\times$ |
| | Llama-3.1-8B$_{\text{INST}}$ | 17.42 | | $4.2\times$ |

## C  IMPACT OF INTERMEDIATE VERIFIER SELECTION

The selection of the intermediate verifier is critical to efficiently navigate the trade-offs between throughput versus acceptance rates. Selecting an intermediate verifier that is too small improves throughput but may not be able to accurately identify *incorrect* draft tokens. In contrast, choosing a large intermediate verifier enables a more accurate rejection of incorrect draft tokens, but reduces throughput, defeating its purpose altogether. To maximize benefits, we must select a sweet-spot on this trade-off curve.

To assess the impact of intermediate verifier selection, we conduct an extensive study using the Llama 3.1 8B model as the target model and the ShareGPT dataset. It consists of 32 decoder layers. We evaluate different draft model size by varying the exit layers they use, sweeping it from Layer 1 to Layer 30. Similarly, we use different intermediate verifiers by sweeping the exit layers from Layer 2 to Layer 31. We analyze the throughput of all these draft and intermediate verifier combinations. Figure 8 shows the relative throughput of each combination compared to vanilla decoding for each draft and intermediate verifier combination.

Our evaluations show that among all configurations, the setting with draft generation at Layer 3 and intermediate verification at Layer 8 achieves the highest throughput. We use this setting as the default in our HiSpec design while using this model. In contrast, the configurations shown in the bottom row, where drafts are always verified only at Layer 31 degrades performance by up to $1.4\times$. This setting approximates Layerskip based speculative decoding. These results highlight that incorporating intermediate verification to discard inaccurate tokens *early*, is critical for improving throughput. Moreover, it is important to select the right intermediate verifier to maximize throughput benefits.

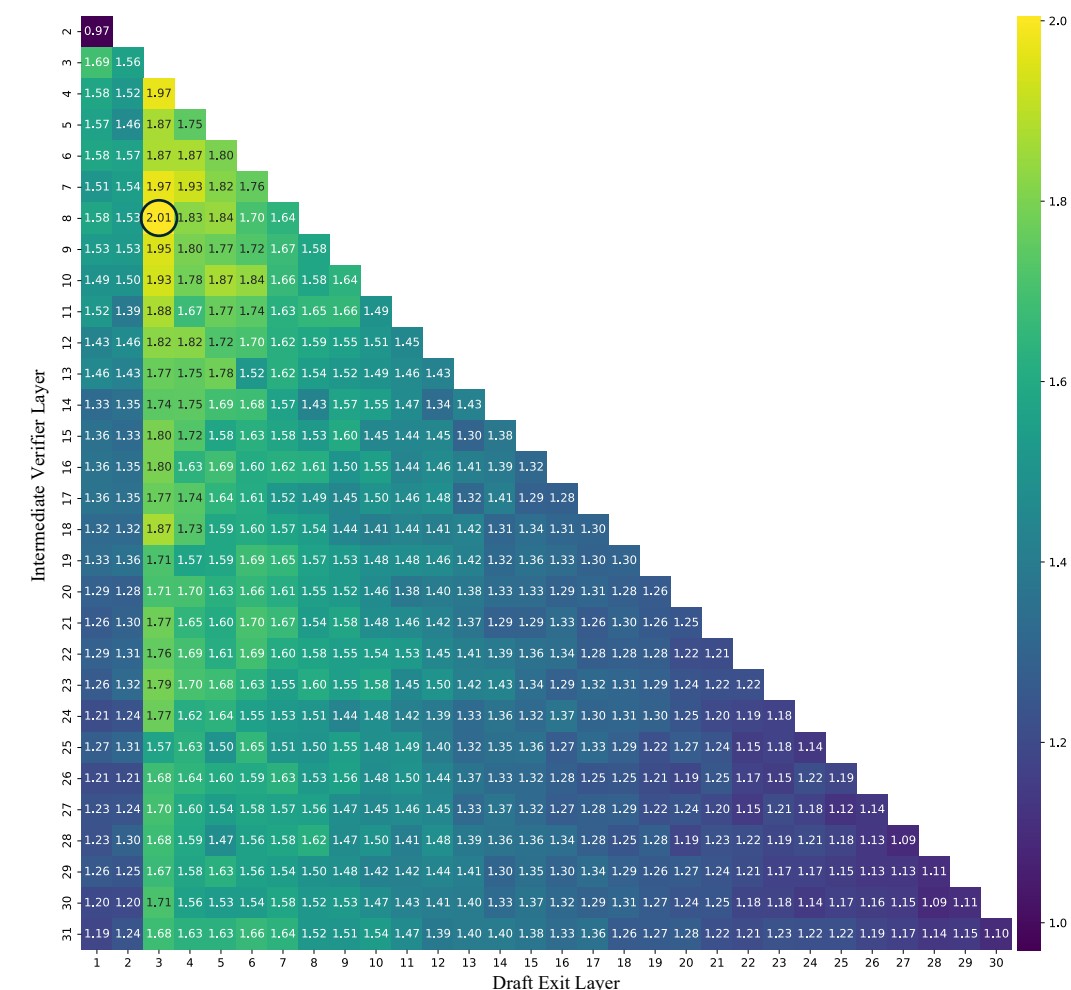

Figure 8: Throughput of all valid combinations of drafts and intermediate verifiers for the 32 layer Llama 3 8B model on the ShareGPT dataset relative to vanilla decoding (*higher is better*).

## D    BENCHMARKS USED FOR EVALUATIONS

We evaluate HiSpec across a wide range of LLM tasks, including dialogue, text summarization, code-generation, and mathematical reasoning, covering a broad spectrum of model capabilities. The remainder of this section details the specific of the benchmarks used in this evaluation. Our selection of benchmarks is consistent with prior works.

**Dialogue:** Drawn from real LLM interactions, ShareGPT sha (2023) is an open-ended dialogue benchmark. The dataset was collected from users voluntarily sharing their GPT conversations, providing natural multi-turn exchanges. These prompts span a wide variety of topics, making the benchmark a realistic test of a model's ability to generate coherent, contextually appropriate responses.

**Text Summarization:** CNN Dailymail (CNN/DM) Nallapati et al. (2016) and Xsum Narayan et al. (2018) are popular summarization benchmarks. Both CNN/DM and Xsum are derived from online news articles. In CNN/DM, each article is paired with brief highlights summarizing the article. In contrast, Xsum provides a single-line summary of each article, emphasizing conciseness.

**Code Generation:** We use the HumanEval Chen et al. (2021) dataset to evaluate the robustness of HiSpec on code-generation tasks. HumanEval consists of 164 python programming problems

of varying difficulty. This benchmark suite assesses the ability of a model to produce functionally correct code.

**Mathematical Reasoning:** The GSM8K Cobbe et al. (2021) benchmark suite contains over 8K grade school-level math word problems designed to assess arithmetic reasoning. Each prompt requires the model to perform step-by-step mathematical operations to arrive at the final solution.

**Input and Output Lengths:** Our selection of benchmarks exhibit substantial variation in both input and output token lengths, as shown in Figure 9. For example, summarization tasks such as CNN/DM, Xsum contain longer input sequences compared to other benchmarks. These variations ensure that our evaluations encompasses a diverse range of input-output configurations.

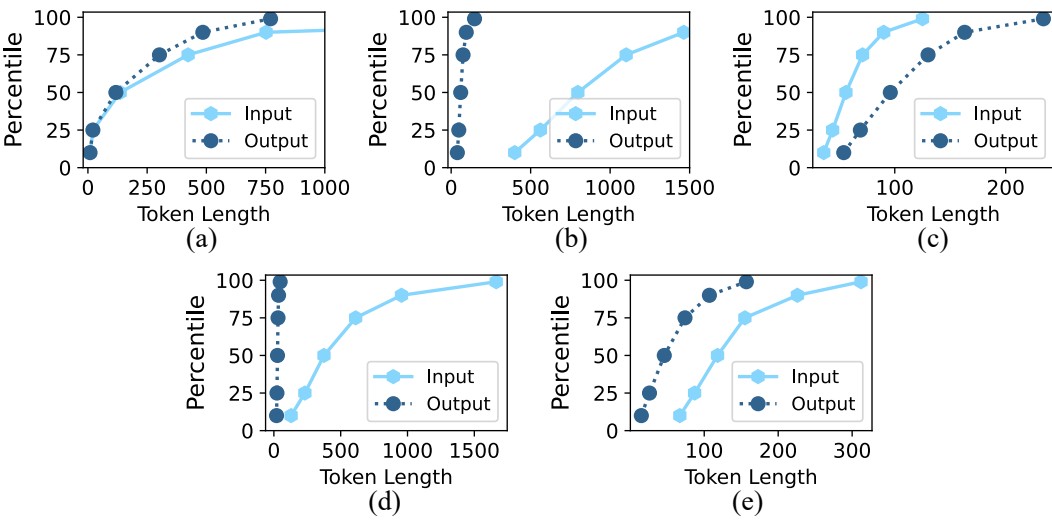

Figure 9: Cumulative distribution of input and output token lengths across (a) ShareGPT, (b) CNN/DM, (c) GSM8K, (d) Xsum, and (e) HumanEval. This distribution reflects how the inherent nature of each task shapes token length. Summarization tasks such as CNN/DM and Xsum contain prompts with higher input token lengths. In contrast, tasks mathematical reasoning tasks such as GSM8K contain much shorter prompts.

## E  RELATED WORK

Prior works improve the efficacy of speculative decoding by accelerating draft generation and target verification. Next, we present an overview of these directions, highlighting representative methods and their key limitations:

**Draft Generation:** Existing approaches that accelerate draft generation can be broadly categorized into methods which optimize the auxiliary draft model to generate accurate tokens or strategies which leverage the target model itself for draft generation. EAGLE Li et al. (2024) designs a draft model to extrapolate the hidden representations of the target model, while SpecInfer Miao et al. (2023) explicitly optimizes the draft model to maximize token acceptance rates. However, these methods necessitate customized training and the reliance on an auxiliary draft model also introduces memory overheads Tang et al. (2025) as the weights and KV caches of the auxiliary draft model need to be stored in GPU memory.

In contrast, methods such as SWIFT Xia et al. (2024) selectively skip layers of the target model itself to efficiently generate draft tokens, but requires substantial workload-specific tuning to isolate layers and operations that can be skipped while maintaining high token acceptance rates. Lookahead decoding Fu et al. (2024) modifies the attention mask to enable the target model to generate multiple tokens in parallel. However, this approach makes each decode step significantly more computationally expensive. Besides these approaches, methods like Layerskip Elhoushi et al. (2024), and

AdaDecode Wei et al. (2025) leverage *early-exit models*, a class of language models which allows tokens to skip layer traversal at selected exit layers. Both these methods exploit this mechanism to employ an exit-layer within the target model for draft generation, and verify these predictions using the remaining target model layers. While Layerskip statically chooses a single draft layer, AdaDecode dynamically decides the draft layer by using confidence thresholds. Although these methods accelerate draft generation, overall throughput is still limited by the verification overhead, particularly for larger models as demonstrated in Appendix B.

These works mainly focus on accelerating draft token generation and cannot address the verification bottleneck, which is significantly more critical.

**Target Verification:** To address the verification bottleneck, SPRINTER Zhong et al. (2025) performs "intermediate verification" using an auxiliary model that discards inaccurate tokens early and invokes the target model only when absolutely necessary. While this improves throughput, SPRINTER incurs substantial trade-offs. Orchestrating the draft, verifier, and the target model simultaneously increases the memory footprint as the weights and KV caches of all participating models need to be stored in GPU memory. Moreover, it introduces overheads to train a verifier model, capable of accurately approximate the target model. SPRINTER also degrades accuracy, as the target model does not validate all intermediate tokens.

Our proposal, HiSpec, addresses the verification bottleneck. HiSpec exploits early-exit models to enable low-overhead intermediate verification using selected layers exit-layers. Moreover, it also uses an exit layer (smaller than the intermediate verifier layer) to generate draft tokens. Unlike prior works that incur training overheads for intermediate verification and draft generation. HiSpec can be naturally applied to any early-exit model (with at least two exits). HiSpec does not introduce any training or memory overheads and maintains the accuracy of the target model.

