# OpenReview forum: "HiSpec: Hierarchical Speculative Decoding for LLMs"
_ICLR.cc/2026/Conference — Submitted to ICLR 2026_

### Official Review · Reviewer_sixV · 2025-10-20

**Soundness:** 2
**Presentation:** 2
**Contribution:** 2
**Rating:** 4
**Confidence:** 4

**Summary:**

The paper proposes HiSpec to speed up LLM inference by inserting a low-overhead intermediate verification step between the draft and target model using early-exit (EE) layers of the target, proposing reusing KV caches and hidden states, and periodic full-model verification.  Several experiments across multiple tasks and model families demonstrate the effectiveness of the method.

**Strengths:**

- This paper is overall well-written.
- Using exit layers avoids training an additional auxiliary verifier and reduces memory complexity.
- The design addresses alignment challenges which may be useful for real systems.

**Weaknesses:**

- Although the paper explains the omission (unavailable verifier models and accuracy costs), the author may want to provide an approximate or partial reproduction to contextualize the gains of SPRINTER.
- The “¼-depth verifier / ⅛-depth drafter” rule of thumb could benefit from a more systematic cross-family analysis (beyond the provided heatmaps) or an adaptive policy.
- The author may want to provide more results using more recent LLMs, including Llama3-70B and Qwen3 series models.
- From a practical perspective, the authors should consider combining HISPEC with existing speculative sampling methods, such as Eagle, to demonstrate the effectiveness of HISPEC.
-  The default Ni=4 is justified by ablations,  which seems like a magic number without other guarantees. The author may need to conduct more exploration of adaptive Ni (e.g., driven by acceptance confidence/entropy) to provide more insight into HISPEC.

**Questions:**

- How does HiSpec behave under high-throughput server settings (many concurrent sequences), especially regarding memory pressure and cache fragmentation when pruning KV for rejected tokens? Any interactions with paged attention/paging strategies?
-  Could Ni and Nd be adapted online for real servers?
-  Beyond Llama/CodeLlama, have you observed similar “¼-depth works best” dynamics on other transformer families (e.g., OPT, Qwen, or Gemma families)? We need some insight rather than just a magic number.

---

> ### Author Response · Authors · 2025-11-23
>
> We thank the reviewer for their feedback. We have addressed the points raised in the review below:
>
> **Q-1) Comparison with SPRINTER**
> R-1) The auxiliary verifier in SPRINTER is tightly coupled to both the target model and the specific task distribution. Consequently, an independent re-implementation would not be a faithful representation of SPRINTER. However, in terms of raw speedup, SPRINTER achieves a maximum of 1.83$\times$ over prior work, while HiSpec improves throughput by up to 2.01$\times$. Furthermore, we note that SPRINTER is a lossy speculation method, as not all intermediate tokens are verified through the target model, while HiSpec is a loss-less method. Table-4 highlights the comparison between SPRINTER and HiSpec.
>
> **Table-4:** Comparison with SPRINTER.
>
> | Feature || SPRINTER | HiSpec (Ours) |
> | :---: | :---: | :---: | :---: |
> | **Max Speedup** || 1.83$\times$ | **2.01$\times$** |
> | **Lossless Generation** || No (Lossy) | **Yes** |
> | **Demonstration Scale** || Small Models ($<1.6$B) | **Large Models (70B)** |
>
> **Q-2) Memory Pressure**
> R-2) HiSpec inherently reduces memory pressure as it re-uses KV cache across the draft, intermediate verifier and the full model, thereby eliminating the overhead and complexity of managing multiple models. Pruning rejected tokens is handled via tensor slicing, which is compatible with standard KV cache allocators. Furthermore, HiSpec conserves memory bandwidth by rejecting inaccurate tokens early, reducing overhead and contention for concurrent sequences.
>
>
> **Q-3) Adapting Ni and Nd in online setting**
> R-3) Yes, Ni and Nd could be adapted in online setting. However, prior work has already explored dynamic speculation lengths by predicting the rejection of draft tokens [1] and using entropy [2]. These approaches address policy selection for the number of speculated tokens and are *complementary* to HiSpec. We focus our analysis to demonstrate the benefits of Hierarchical speculation.
>
> **Q-4) Applicability on Other Families**
> R-4) The optimal placement of the intermediate verifier depends on the depth at which the hidden states stabilize enough to allow effective generation of draft tokens. Prior work shows that shallow semantic features stabilize at 1/4th the model depth [3]. Our proposed guideline is only intended to assist the configuration of HiSpec in real-world serving scenarios.
>
> To demonstrate the flexibility of HiSpec on other choices of model families and intermediate verifier, we fine-tune two OPT models to first enable early exit functionality with exits (auxiliary heads) at 1/4th depth intervals. We request the reviewer to consult **R-2** in the response to **Reviewer ZW4r** for further details. In the experiments shown in Table-1, both OPT-6.7B and 1.3B models are configured to use the first head (1/4th depth) for draft generation and second head (1/2 depth) for intermediate verification. While the absence of pre-trained early-exit checkpoints prevents fine-grained exploration of the optimal draft and intermediate verifier layer for other families such as Qwen, our results on OPT models show that the speedup benefits of HiSpec are *robustly transferable* to other architectures.
>
> $\tiny[1] \text{Huang, Kaixuan, Xudong Guo, and Mengdi Wang. "SpecDec++: Boosting Speculative Decoding via Adaptive Candidate Lengths." Workshop on Efficient Systems for Foundation Models II@ ICML2024.}$
> $\tiny[2] \text{Mamou, Jonathan, et al. "Dynamic Speculation Lookahead Accelerates Speculative Decoding of Large Language Models." NeurIPS Efficient Natural Language and Speech Processing Workshop. PMLR, 2024.}$
> $\tiny[3] \text{Jawahar, Ganesh, Benoît Sagot, and Djamé Seddah. "What does BERT learn about the structure of language?." ACL 2019-57th Annual Meeting of the Association for Computational Linguistics. 2019.}$

---

> > ### Comment · Reviewer_sixV · 2025-11-26
> >
> > I thank the authors for their detailed response. However, my concerns have not been adequately addressed.
> > 1. The Model Family Issue
> > The OPT family is now considered relatively dated. Given that the model weights for the Qwen and Llama series are publicly available, identifying auxiliary heads at $x$-th depth intervals is technically straightforward. Furthermore, if the concern regards the availability of matching pre-training data, the OLMoE [1] series has released both its weights and pre-training corpus. I believe that demonstrating results on currently popular, state-of-the-art models would possess significantly greater practical value.
> >
> > [1] OLMoE: Open Mixture-of-Experts Language Models.
> >
> > 2. Issue regarding Ni
> > It still seems that Ni remains a "magic number." This raises significant questions regarding the method's generalizability across models with different architectures or belonging to different families. Additionally, regarding the discussion on the on-line setting, I maintain that empirical experiments are necessary to substantiate your claims rather than relying on theoretical justification alone.
> >
> > Consequently, I believe HiSpec requires major revisions in its current form to address these issues. I am inclined to maintain my current score.

---

> ### Author Response · Authors · 2025-12-03
>
> $\underline{\mathbf{Model\ Family\ Issue}}$
>
> We highlight that our evaluations *already* demonstrate the generalizability of HiSpec on Llama 3, Llama 2 and CodeLlama families (Please refer to Table-1 in the main paper). Furthermore, we selected the OPT family to address the reviewer’s *own explicit request* (OPT, Qwen or Gemma) in the initial review.
>
> Pre-training the OLMoE models to support early-exits (similar to Layerskip) is *non-trivial* because the original training recipe necessitates processing of 5T tokens and originally required 61,000+ H100 GPU-hours (cost > \$150,000) for convergence. Consequently, replicating such a resource-intensive training is beyond the scope of this work.
>
> However, to dispel any concerns regarding the robustness of HiSpec, we further demonstrate HiSpec on the Llama 3.2 1B model. Table-1 shows that the benefits of HiSpec are *easily transferable*.
>
> **Table-1:** Throughput of HiSpec against Vanilla auto-regression decoding across benchmarks on the NVIDIA H100  GPU.
>
> | Metric || ShareGPT | CNN/DM | HumanEval | GSM8K | Xsum |
> |----------|-|----------------|--------------|-----------------|------------|---------|
> | Auto-regressive (tokens/sec) || 91.72   | 91.12     | 91.33    | 91.69   |   90.16   |
> | HiSpec (tokens/sec)               || 137.46 |  143.28  | 142.74  | 144.88 | 132.05   |
> | Speedup  || 1.50$\times$ | 1.57$\times$ | 1.56$\times$ | 1.58$\times$ | 1.46$\times$ |
>
> ---
> ---
>
> **Summary:** Our evaluations now encompass the Llama 2, Llama 3, Llama 3.2, CodeLlama, and OPT families, spanning model sizes from 1B to 70B. This comprehensive coverage surpasses that of state-of-the-art prior works [1-4].
>
> ---
> ---
>
> $\underline{\mathbf{Choice\ of\ Ni}}$
>
> Ablation derived default hyper-parameters (such as Ni) is a *standard* practice in prior work [1-4]. For example, Lookahead decoding [3] relies on fixed window size and n-gram depth rather than dynamic resizing in every step. Similarly, AdaDecode [1] and SWIFT [2] employ exit thresholds and layer configurations respectively which are fixed during inference. Our choice of $N_i=4$ follows this *established methodology* and is *rigorously justified* by our ablations to maximize the trade-off between computational cost and overall performance.
>
> To address the inquiry regarding online adaptation, we implement a lightweight adaptive policy (shown below) that adjusts $N_i$ based on the final acceptance rate. Table-2 shows the performance of this adaptive policy in comparison to static selection of $N_i$.
>
> Formally, let $\alpha_t$ be the final acceptance rate at step $t$. We define a high ($\tau_{high}$) and a low ($\tau_{low}$) threshold to either aggressively increase or conservatively reduce $N_i$ respectively. The value of $N_i$ for the next step, $N_i^{(t+1)}$, is updated according to the following rule:
>
> $$
> N_i^{(t+1)} = \begin{cases}
> \min(N_{\text{max}}, N_i^{(t)} + 1) & \text{if } \alpha_t \ge \tau_{\text{high}} \\\\
> \max(N_{\text{min}}, N_i^{(t)} - 1) & \text{if } \alpha_t < \tau_{\text{low}} \\\\
> N_i^{(t)} & \text{otherwise}
> \end{cases}
> $$
>
> For the results presented in Table 2, we set the thresholds to $\tau_{high}=0.8$ and $\tau_{low}=0.3$, with bounds $N_{\text{min}}=2$ and $N_{\text{max}}=8$. Consequently, the empirical results below validate that HiSpec is not reliant on a static *magic number*, but rather serves as a robust framework that is *fully complementary and further enhanced by* online adaptive strategies.
>
> **Table-2:** Throughput improvement of HiSpec with a dynamic policy (initialized to $N_i=4$) relative to a fixed $N_i=4$
>
> | Model            ||   ShareGPT | CNN/DM  |
> |------------------|-|-----------------|---------------|
> | Llama2 13B  ||       3.0%         |     4.2%    |
> | Llama2 70B  ||       3.7%      |     2.6%    |
>
> ---
> ---
>
> **Summary:** We clarify that $N_i$​ is an empirically optimized hyper-parameter consistent with prior work [1-4]. Furthermore, we perform additional experiments with a dynamic policy to demonstrate that HiSpec is *not* reliant on a magic number and generalizes effectively to online settings.
>
> ---
> ---
>
> As the mentioned concerns have now been fully addressed, we request a reconsideration of the score.
>
> $\tiny[1]  \text{ Wei, Zhepei, et al. "AdaDecode: Accelerating LLM Decoding with Adaptive Layer Parallelism." Forty-second International Conference on Machine Learning (ICML 2025).}$
> $\tiny[2]  \text{ Xia, Heming, et al. "SWIFT: On-the-Fly Self-Speculative Decoding for LLM Inference Acceleration." The Thirteenth International Conference on Learning Representations (ICLR 2025).}$
> $\tiny[3]  \text{ Fu, Yichao, et al. "Break the Sequential Dependency of LLM Inference Using Lookahead Decoding." Forty-first International Conference on Machine Learning (ICML 2024).}$
> $\tiny[4]  \text{ Elhoushi, Mostafa, et al. "Layerskip: Enabling early exit inference and self-speculative decoding." 62nd Annual Meeting of the Association for Computational Linguistics, 2024.}$

---

### Official Review · Reviewer_cjaU · 2025-10-26

**Soundness:** 2
**Presentation:** 3
**Contribution:** 3
**Rating:** 4
**Confidence:** 4

**Summary:**

This paper proposes HiSpec, a hierarchical early-exit speculative decoding method leveraging middle-layer exit as intermediate verifier. The paper reveals that the bottleneck of speculative decoding lies in verification stage, and intermediate verification (before final verification) can largely mitigate this bottleneck. Therefore, HiSpec leverages middle-layer early exit for intermediate verification, creating a hierarchical speculation framework. Furthermore, the hidden states from early-exit drafting can be re-used in verification, avoiding re-computation.

**Strengths:**

The topic is highly related to practical issues of LLM acceleration. The observation of speed gap between drafting and verification is important, and essential in optimizations of current research.

HiSpec involves no training overheads. It leverages existing early-exit checkpoints from Layerskip for early-exit drafting and middle-layer intermediate verification.

The ablation studies in fig.6 and fig.7 about experimental configurations (e.g. speculation lengths) are comprehensive and empirically convincing.

**Weaknesses:**

The accuracy of intermediate verification can largely affect the overall performances, while the paper only provided end-to-end speed, but no speculation accuracies. The accuracy results would further demonstrate the effectiveness of the method.

The configurations of baselines are not sufficiently specified and tuned. While HiSpec uses 1/8 layers as drafter and 1/4 layers as intermediate verifier, the exit layer of LayerSkip, and the configuration for LookAhead should be specified. Moreover, these configs for baselines should be tuned for optimal performances.

The usage is limited to already-trained early-exit models like Layerskip. The paper claims that HiSpec can also be applied to post-trained models, but provides no empirical evidence for it.

There are some other techniques to improve final verification accuracy, e.g. tree attention, while this paper adopts none of them. It is unclear whether the effectiveness of intermediate verification still preserves when combined with these techniques, as the false rejection of intermediate verification may outweigh the benefits when the final accuracy is high.

The paper can be better organized. The ‘method’ section should focus more on the overall design, while the experimental details (e.g. the exit layer) should be put to the ‘experiment’ section.

**Questions:**

1. What is the speculation accuracy of intermediate verification? How is it compared to final verification?
2. Can you provide more detailed configurations of baselines, such as the exit layer of Layerskip and the configs of LookAhead? Can you tune these hyper-parameters and report the optimal performances?
3. Can you provide evidences of the wider applicability to post-trained models?

---

> ### Author Response · Authors · 2025-11-23
>
> We thank the reviewer for their feedback. We have addressed the points raised in the review below:
>
> **Q-1) Acceptance Rate at Intermediate and Final Layer**
> R-1) We focus on throughput for our evaluations because acceptance rate (or speculation accuracy) is agnostic to computational overhead. We request the reviewer to consult **R-3** in the response to **Reviewer ZW4r** for further information. However, we will add a section in the paper to summarize acceptance rates.
>
> Instead of simply relying on a single exit layer for draft generation, HiSpec simultaneous leverages an intermediate layer to act *both* as a (less precise) verifier, and a (more accurate) draft layer. This allows HiSpec to signficantly improve speculation accuracy. Table-2 summarizes the acceptance rate of baseline single EE speculation, and at intermediate and final layer using HiSpec for Codellama-34B model across various benchmarks.
>
> **Table-2:** Comparison of Baseline (single EE draft directly verified by the full model) against HiSpec, showing acceptance rate at both the intermediate and full model verification stages across benchmarks for the Codellama-34B model.
>
> | Method | ShareGPT | CNN/DM | HumanEval | GSM8K | Xsum |
> | :--- | :---: | :---: | :---: | :---: | :---: |
> | Single EE Speculation (Baseline) | 39% | 19% | 42% | 35% | 23% |
> | Intermediate Verification | 77% | 70% | 80% | 81% | 70% |
> | Full Model Verification | 57% | 70% | 62% | 57% | 53% |
>
> **Q-2) Baseline Configuration of other Methods**
> R-2) We will add a section to summarize these details in the main paper:
> * LookAhead Decoding: We perform a search to empirically determine the optimal hyper-parameters to be consistent with evaluations in prior work [1].
>
> * Layerskip: To maintain consistency, we adopt the configuration reported in the Layerskip paper [2] when evaluating the same model and benchmark configuration. When the configuration is unavailable, we perform a search to empirically determine the optimal hyper-parameters. For example, Table-3 shows that for the Llama2-70B model, the best throughput is achieved by using the 20th layer (1/4th depth) as the draft and speculating eight tokens per step.
>
> **Table-3:** Throughput (tokens/sec) with Layerskip by varying exit layer and number of speculations for the CNN/DM summarization dataset on 2$\times$ NVIDIA H100 GPUs.
>
> | **Exit Layer** | | **Num Speculation** | | |
> | :---: | :---: | :---: | :---: | :---: |
> | | **6** | **8** | **12** | **14** |
> | 10 | 22.93 | 22.21 | 17.41 | 17.30 |
> | 16 | 23.85 | 23.57 | 21.32 | 19.08 |
> | 20 | 24.29 | **24.47** | 23.46 | 20.80 |
> | 24 | 22.95 | 23.92 | 22.24 | 19.58 |
>
> **Q-3) Applicability to Post-Training Modified Methods**
> R-3) To demonstrate the applicability of HiSpec with post-training modified methods, we add auxiliary heads on the OPT 6.7B and 1.3B models and fine-tune this model for 50K iterations on the Pile dataset. Next, we implement HiSpec for these models. We request the reviewer to consult **Table-1** in the response to **Reviewer ZW4r** for further detauils. This demonstrates that HiSpec *seamlessly generalizes* to post-training models.
>
> **Q-4) Compatibility with Tree Attention**
> R-4) HiSpec decreases verification latency by re-purposing intermediate layers for partial verification. While our approach is broadly applicable to any speculative method, it would be *particularly advantageous* for methods which use Tree Attention. This is because while tree-attention improves acceptance rate, it incurs *significant verification overhead* [3,4] as it necessitates the verification of a large number of draft tokens.
>
> Integrating HiSpec with Tree Attention presents significant complexity as KV cache and hidden states would need to be buffered and managed across multiple speculation paths per request at intermediate layers. This mechanism is intricate and non-trivial, consequently, we defer this to future work.
>
> **Q-5) Editorial Feedback**
> R-5) Thankyou, we will reorganize the paper. Our current organization was structured to use motivational data to explain the default design choices in HiSpec.
>
> $\tiny[1] \text{Wei, Zhepei, et al. "AdaDecode: Accelerating LLM Decoding with Adaptive Layer Parallelism." Forty-second International Conference on Machine Learning (ICML), 2025.}$
> $\tiny[2] \text{Elhoushi, Mostafa, et al. "LayerSkip: Enabling Early Exit Inference and Self-Speculative Decoding." ACL (1). 2024.}$
> $\tiny[3] \text{Yao, Jinwei, et al. "DeFT: Decoding with Flash Tree-attention for Efficient Tree-structured LLM Inference." The Thirteenth International Conference on Learning Representations (ICLR), 2025.}$
> $\tiny[4] \text{Wang, Jikai, et al. "OPT-Tree: Speculative Decoding with Adaptive Draft Tree Structure." Trans. Assoc. Comput. Linguistics (TACL), 2025.}$

---

> > ### Comment · Reviewer_cjaU · 2025-11-24
> >
> > Thank you for preparing the rebuttal. I am looking forward to some further explanations about Quenstion 2:
> >
> > By raising Question 2, I want to know that: did you report the optimal inference speed of all methods, with careful tuning on configurations, in Table1?
> >
> > If so, please provide the optimal configurations for each combination of **(method, model type, dataset)** , e.g. exiting layer and speculation length. And also the optimal intermediate layer, for HiSpec. I believe these data is quite essential for the reproductability and effectiveness.
> >
> > The idea of intermediate verification is highly empirical, but the configurations are missing. If additional information or proper justification can be provided, I will consider upgrading this paper.

---

> > > ### Author Response · Authors · 2025-12-04
> > >
> > > ## $\underline{\text{Configurations\ Of\ Baseline\ Methods}}$
> > >
> > > Yes, our evaluations are consistent with the prior state-of-the-art work [1], performance for baseline methods is tuned and representative. The hyper-parameter configuration for each prior work is detailed below:
> > >
> > > ### **$\underline{\text{Layerskip}}$**
> > > Layerskip is parameterized by exit layer ($L_{ext}$) and the number of speculations ($N_{spec}$). To maintain consistency, we adopt the configuration reported in the Layerskip paper [2] when evaluating the same model and benchmark configuration. When the configuration is unavailable, we perform a search to empirically determine the optimal hyper-parameters (summarized in Table-1).
> > >
> > > **Table 1:** Optimal LayerSkip configurations $(L_{ext}, N_{spec})$, where $L_{ext}$ is the exit layer and $N_{spec}$ is the number of speculated tokens.
> > >
> > > | **Model** | **ShareGPT** | **CNN/DM** | **HumanEval** | **GSM8K** | **XSum** |
> > > | :--- | :---: | :---: | :---: | :---: | :---: |
> > > | **Llama3-8B**          | $(8, 8)$       | $(8, 12)$    | $(8, 4)$       | $(6, 6)$      | $(10, 6)$  |
> > > | **Llama2-7B**          | $(8, 4)$       | $(8, 12)$    | $(8, 6)$       | $(8, 6)$      | $(8, 12)$  |
> > > | **Llama2-13B**        | $(15, 4)$     | $(15, 12)$  | $(7, 4)$       | $(10, 4)$    | $(15, 4)$  |
> > > | **CodeLlama-7B**   | $(10, 4)$     | $(12, 8)$    | $(8, 6)$       | $(8, 6)$       | $(12, 6)$  |
> > > | **CodeLlama-34B** | $(8, 12)$     | $(12, 12)$  | $(12, 6)$     | $(12, 4)$     | $(16, 4)$  |
> > > | **Llama2-70B**        | $(12, 6)$     | $(20, 8)$    | $(20,6)$      | $(16, 4)$     | $(20, 6)$  |
> > >
> > > ### **$\underline{\text{LookAhead\ Decoding}}$**
> > > Lookahead Decoding contains three hyper-parameters: window size ($W$), n-gram size ($N$), and the guess set size ($G$). We perform a search to empirically determine the optimal hyper-parameters (summarized in Table-2).
> > >
> > > **Table 2:** Optimal Lookahead configurations $(W, N, G)$, where $W$ is the window size, $N$ is the $n$-gram size, and $G$ is the guess set size.
> > >
> > > | **Model** | **ShareGPT** | **CNN/DM** | **HumanEval** | **GSM8K** | **XSum** |
> > > | :--- | :---: | :---: | :---: | :---: | :---: |
> > > | **Llama3-8B**          | $(5, 7, 7)$    | $(3, 10, 10)$ | $(5, 8, 8)$     | $(6, 6, 6)$     | $(4, 8, 8)$     |
> > > | **Llama2-7B**          | $(5, 8, 8)$    | $(6, 4, 4)$     | $(5, 5, 5)$     | $(3, 10, 10)$ | $(4, 10, 10)$ |
> > > | **Llama2-13B**        | $(4, 4, 4)$    | $(3, 10, 10)$ | $(5, 8, 8)$     | $(4, 4, 4)$     | $(3, 10, 10)$ |
> > > | **CodeLlama-7B**   | $(3, 10, 10)$ | $(4, 8, 8)$     | $(4, 10, 10)$ | $(5, 7, 4)$    | $(5, 7, 7)$     |
> > > | **CodeLlama-34B** | $(3, 10, 10)$ | $(5, 5, 5)$     | $(4, 10, 10)$ | $(4, 8, 8)$    | $(3, 10, 10)$  |
> > >
> > > ### **$\underline{\text{AdaDecode}}$**
> > > AdaDecode is parameterized by confidence threshold ($\gamma$) and the number of speculations ($N_{spec}$). Following the methodology used for prior methods, we perform an empirical search to identify the optimal hyper-parameters (summarized in Table-3).
> > >
> > > **Table 3:** Optimal AdaDecode configurations $(\gamma, N_{spec})$, where $\gamma$ is the confidence threshold and $N_{spec}$ is the number of speculated tokens.
> > >
> > > | **Model** | **ShareGPT** | **CNN/DM** | **HumanEval** | **GSM8K** | **XSum** |
> > > | :--- | :---: | :---: | :---: | :---: | :---: |
> > > | **Llama3-8B**          | $(0.75, 6)$ | $(0.75, 8)$ | $(0.75, 4)$ | $(0.75, 6)$  | $(0.75, 6)$ |
> > > | **CodeLlama-34B** | $(0.75, 4)$ | $(0.75, 6)$ | $(0.75, 6)$ | $(0.75, 6)$  | $(0.75, 6)$ |
> > >
> > > ### **$\underline{\text{SWIFT}}$**
> > > SWIFT is parametrized by arrays for the skipped layers (both attention and MLP). Following prior work [1,3], we run a Bayesian optimization search for 200 iterations to determine skipped layers for configuring the drafter model. We highlight the configuration for the CNN/DM dataset as a representative exemplar:
> > >
> > > | Model | Skipped Attention Layers | Skipped MLP Layers |
> > > | :--- | :--- | :--- |
> > > | **Llama2-13B** | 2, 5, 6, 10, 11, 14, 15, 20, 22, 23, 24, 25, 26, 27, 28, 30, 32, 33, 34, 35 | 9, 11, 14, 15, 19, 20, 24, 25, 26, 27, 28, 29, 30, 31 |
> > > | **CodeLlama-34B** | 9, 10, 12, 19, 20, 22, 24, 27, 29, 31, 32, 33, 34, 35, 36, 37, 38, 40, 41, 42, 46 | 6, 8, 9, 10, 13, 16, 18, 19, 20, 22, 26, 27, 30, 31, 32, 33, 36, 38, 41, 43, 44 |
> > > | **Llama2-70B** | 8, 9, 14, 15, 17, 18, 19, 21, 22, 25, 26, 27, 28, 29, 30, 31, 32, 35, 36, 38, 39, 43, 45, 48, 51, 53, 54, 56, 57, 61, 62, 63, 65, 66, 69, 70, 71, 73, 74, 77, 78 | 3, 9, 12, 13, 14, 15, 16, 18, 22, 26, 27, 29, 30, 39, 41, 42, 44, 46, 49, 50, 51, 52, 53, 54, 55, 56, 57, 60, 62, 63, 64, 66, 69, 71, 72, 75, 76 |
> > >
> > > $\tiny[1]  \text{Wei, Zhepei, et al. "AdaDecode: Accelerating LLM Decoding with Adaptive Layer Parallelism." ICML 2025}$
> > > $\tiny[2]  \text{Elhoushi, Mostafa, et al. "Layerskip: Enabling early exit inference and self-speculative decoding." ACL, 2024.}$
> > > $\tiny[3]  \text{Zhang, Jun, et al. "Draft and verify: Lossless large language model acceleration via self-speculative decoding." ACL, 2024.}$

---

> ### Author Response · Authors · 2025-12-04
>
> ## $\underline{\text{Optimal\ Intermediate\ Layer\ for\ HiSpec}}$
>
> While the main paper notes that the optimal intermediate layer generally lies near $1/4^{th}$ the model depth, we agree that providing the specific value adds clarity. Therefore, we have explicitly listed the optimal intermediate layer across all evaluated benchmarks and models in Table-5.
>
> **Table 5:** Optimal Intermediate Layer for HiSpec across all evaluated models and benchmarks.
>
> | **Model** | **ShareGPT** | **CNN/DM** | **HumanEval** | **GSM8K** | **XSum** |
> | :--- | :---: | :---: | :---: | :---: | :---: |
> | **Llama3-8B** (32 Layers)           |  $8$   |  $10$   |  $8$   |  $8$   |  $9$   |
> | **Llama2-7B** (32 Layers)           |   $8$   |  $8$   |  $8$   |  $8$   |  $8$  |
> | **Llama2-13B** (40 Layers)         | $10$    |  $15$   | $12$    |  $10$   |  $10$   |
> | **CodeLlama-7B** (32 Layers)    |  $8$   |  $12$   |  $8$   |  $8$   | $8$   |
> | **CodeLlama-34B** (48 Layers)  |  $12$   |  $16$   |  $12$   |  $12$   |  $12$   |
> | **Llama2-70B**  (80 Layers)        |  $20$   |  $24$   |  $20$   |  $19$   |  $20$   |

---

### Official Review · Reviewer_ZW4r · 2025-11-01

**Soundness:** 3
**Presentation:** 2
**Contribution:** 2
**Rating:** 4
**Confidence:** 4

**Summary:**

HiSpec introduces Hierarchical Speculative Decoding (HiSpec) — a framework that employs Early-Exit (EE) models to perform low-overhead intermediate verification within the target model itself. The authors observed that the verification step is up to 10× slower than draft generation in typical speculative decoding. Thus, they proposed to use EE models to perform drafting and verification at shallower model layers. HiSpec manages the KV cache dynamically and share the KV cache across the drafter, intermediate verifier and full verifier to reduce memory footprint.

**Strengths:**

- This paper presents a novel way to deal with verification cost, it is among the first few to target the verification wall effectively.
- The motivation is well stated and supported by data
- The idea to reuse early-exit checkpoints as hierarchical verifiers is good — no extra training, minimal overhead.

**Weaknesses:**

- line 362: hug -> hub
- This paper selects the baselines which cut down the draft generation time and discards the comparison with the verification-focused methods, which makes its evaluation incomplete
- This paper assumes that EE checkpoints are available, which are not applicable for all LLMs; adapting HiSpec to vanilla models might need further training.

**Questions:**

- while one-fourth of the model is sufficient to generate up to 69% of the output tokens correctly, Figure 4 shows that for many tasks, the accuracy is well below 50%, which is quite low. How does this accuracy affect the final speedup?
- How are the acceptance lengths/rates like as they are not presented in the experiments?
- can you explain why is it hierachical?

---

> ### Author Response · Authors · 2025-11-23
>
> We thank the reviewer for their feedback. We have addressed the points raised in the review below:
>
> **Q-1) Comparison with Verification Methods:**
> R-1) We found SPRINTER as the only explicit verification focused approach. However, SPRINTER is a lossy-speculation method as it does not validate all intermediate tokens with the full model. Furthermore, it requires the training of an auxiliary model which is tightly coupled to the target model and the specific task distribution. Consequently, an independent reimplementation risks suboptimal convergence and would not be a faithful representation. All other methods [EAGLE, Medusa etc] are not verification-focused.
>
> **Q-2) Generalizability of HiSpec**
> R-2) There are two steps to apply our approach to an existing model:
> * *Step-1*: Adding Early-Exit (EE) functionality
> * *Step-2*: Implement HiSpec on top of the modified model
>
> Step-2 is the primary focus of our paper, given that Step-1 has been thoroughly addressed in prior work [1,2]. Typically, adding EE functionality to an off-the-shelf model requires the addition and fine-tuning of auxiliary heads at selected exit layers using the model’s pre-training data. For example, we fine-tune the OPT 6.7B and 1.3B models on the Pile dataset, which was used during its original pre-training. Table-1 shows that the benefits of HiSpec are *easily transferable* to other models.
>
> **Table-1:** Throughput with HiSpec against Vanilla auto-regressive generation for the CNN/DM and HumanEval dataset on NVIDIA H100.
>
> | Models   ||  |  CNN/DM  |  ||  | HumanEval | |
> |--|-|-|--|--|-|--|--|--|
> |                   || Vanilla | HiSpec | Speedup || Vanilla | HiSpec  | Speedup |
> | OPT 6.7B  || 51.42   | 100.87 | 1.96         || 50.64  | 85.94     | 1.70        |
> | OPT 1.3B  || 78.57   | 141.00 | 1.79         || 77.65  | 112.54   | 1.80        |
>
> When the original pre-training data is not accessible, EE functionality can still be enabled by generating synthetic data directly from the model [3]. However, implementing EE using synthetic data is beyond the scope of this work.
>
> **Q-3) Acceptance Rates with HiSpec**
> R-3) There is an inherent tradeoff between the acceptance rate at the intermediate verifier and the associated computational overhead. While using a deeper layer for intermediate verification would improve acceptance rate, the increased computational cost diminishes the overall throughput benefits. Therefore, even with acceptance rates below 50% for some tasks, the system achieves a net speedup because the cost of generating intermediate tokens is computationally inexpensive. We request the reviewer to consult **R-1** in the response to **Reviewer cjaU** for acceptance rates.
>
>
> **Q-4) Hierarchical Terminology**
> R-4) HiSpec is termed Hierarchical because it inserts an intermediate verification stage between the draft and the full model. This creates a *hierarchy* where tokens are first proposed by the draft, then tentatively verified by the intermediate verifier layer and finally accepted by the full model. To ensure this multi-level speculation remains computationally efficient, HiSpec re-uses KV cache between the draft, the intermediate verifier and the full model.
>
> $\tiny[1]  \text{Chen, Yanxi, et al. "EE-LLM: Large-Scale Training and Inference of Early-Exit Large Language Models with 3D Parallelism." Forty-first International Conference on Machine Learning (ICML), 2024.}$
> $\tiny[2] \text{Pan, Xuchen, et al. "EE-Tuning: An Economical yet Scalable Solution for Tuning Early-Exit Large Language Models." CoRR (2024).}$
> $\tiny[3] \text{Valade, Florian. "Accelerating Large Language Model Inference with Self-Supervised Early Exits." arXiv preprint arXiv:2407.21082 (2024).}$

---

### Author Response · Authors · 2025-12-04
**Summary of Rebuttal Discussion for Area Chair**

Dear Area Chair,

We thank the reviewers for their thoughtful feedback and questions. To assist with changes made to the review process, we provide a summary of (1) our work’s contributions and (2) our responses to reviewer concerns. We've linked the detailed responses for convenience.

## **Summary of Contributions**
HiSpec contributes the following core advances to LLM inference acceleration:
* Identifies long verification latencies as a critical bottleneck to speculative decoding throughput. Most prior works speed up draft generation but do not address the verification overhead.
* Proposes *Hierarchical Speculation Decoding*, a novel framework which addresses verification bottleneck by leveraging intermediate layers in early-exit (EE) models to perform low-overhead intermediate verification so that inaccurate draft tokens are rejected early.
* Facilitates re-use of key-value (KV) cache and hidden states across the draft, intermediate verifier and target, to improve resource-efficiency and reduce computational-redundancy.
* Ensures output response consistency with the full-model by performing periodic target (full-model) verification. In contrast, prior verification-focused methods (SPRINTER) sacrifice output parity by skipping target model verification for all intermediate tokens.

HiSpec is an important first step toward verification-aware speculative decoding. It complements existing frameworks by serving as a modular enhancement for strategies constrained by high-verification overheads (such as *Tree-Attention* based methods).

## **Summary of Responses**
| Reviewer Concern | Response Summary |
| --- | --- |
| [Applicability on Post-Training Modified Models](https://openreview.net/forum?id=CYGI23WQjI&noteId=wrUvnlSMDa) (Reviewer ZW4r, cjaU) | To demonstrate the applicability of HiSpec on post-training modified models, we add auxiliary heads on the OPT 6.7B and 1.3B models and fine-tune this model for 50K iterations on the Pile dataset. Next, we implement HiSpec for these models to demonstrate that HiSpec *seamlessly generalizes* to post-training methods.  |
| [Speculation Accuracy](https://openreview.net/forum?id=CYGI23WQjI&noteId=WQKl6h7tms) (Reviewer ZW4r, cjaU) | Instead of simply relying on a single exit layer for draft generation, HiSpec simultaneously leverages an intermediate layer to act both as a (less precise) verifier, and a (more accurate) draft layer. This enables HiSpec to significantly *improve speculation accuracy* as detailed in the linked response. |
| [Robustness of Baseline Methods](https://openreview.net/forum?id=CYGI23WQjI&noteId=2h0W3i5oN0) (Reviewer cjaU) | Our evaluations are consistent with the prior works, *performance for all baseline methods is tuned and representative*. The hyper-parameter configuration for each prior work is detailed in the linked response. |
| [Comparison with SPRINTER](https://openreview.net/forum?id=CYGI23WQjI&noteId=8dvyxYkwzc) (Reviewer ZW4r, sixV) |The auxiliary verifier in SPRINTER is tightly coupled to both the target model and the specific task distribution. Consequently, an independent re-implementation would not be a faithful representation of SPRINTER. However, in terms of raw speedup, SPRINTER achieves a maximum of 1.83$\times$ over prior work, while *HiSpec improves throughput by up to 2.01$\times$*. Furthermore, we note that SPRINTER is a lossy speculation method, as not all intermediate tokens are verified through the target model, while *HiSpec is a loss-less* method.  |
| [Adapting Speculation Length ($N_i$) Online](https://openreview.net/forum?id=CYGI23WQjI&noteId=aauO5sw0jJ) (Reviewer sixV) | We implement a lightweight adaptive policy that adjusts speculation length based on the final acceptance rate. This dynamic policy achieves a performance *improvement of upto 4.2%* over the static baseline. These results validate that HiSpec is a robust framework that is *fully complementary and further enhanced by* online adaptive strategies. |
| [Generalizability across Model Families](https://openreview.net/forum?id=CYGI23WQjI&noteId=aauO5sw0jJ) (Reviewer sixV) | In addition to our original results on Llama 3, Llama 2 and CodeLlama families, we extend our evaluations to include the OPT and Llama3.2 families. HiSpec is now validated across *five distinct LLM families, spanning model sizes from 1B to 70B*. This comprehensive coverage surpasses that of state-of-the-art prior works. |
| [Availability of Early-Exit (EE) Checkpoints](https://openreview.net/forum?id=CYGI23WQjI&noteId=wrUvnlSMDa) (Reviewer ZW4r) | HiSpec assumes the availability of EE checkpoints, as enabling EE on off-the-shelf models is *thoroughly studied in prior work* [1,2]. Typically, this requires the addition and fine-tuning of auxiliary heads at selected exit-layers using the model's pre-training data. We demonstrate the feasibility of this setup by fine-tuning the OPT models on the Pile dataset. More details have been presented in the linked response. |

---

> ### Author Response · Authors · 2025-12-04
>
> ## **Final Remarks**
> Reviewers acknowledged HiSpec as a *timely and significant contribution* to improving speculative decoding throughput, specifically noting its novel approach to mitigating the verification bottleneck via *lightweight intermediate verification (training-free) and its resource-efficient design that maximizes KV cache reuse*. The primary concerns centered on applicability to post-training methods, robustness of baseline methods, and the rigidity of speculation length ($N_i$​). Our response addresses these by adding and fine-tuning auxiliary heads on the OPT model, detailing the optimal configurations for baseline methods, and implementing a new online adaptive policy that dynamically updates $N_i$ based on acceptance rate. To further demonstrate generalizability, we extend our evaluations to include the Llama 3.2 family, broadening our coverage to encompass five model families spanning 1B to 70B parameters. We believe these additions substantially strengthen the paper’s claims on robustness and broad applicability. We thank the reviewers and Area Chair for their thoughtful feedback and time.
>
> Kind Regards,
> The Authors
>
> ---
> ---
>
> $\tiny[1]  \text{Chen, Yanxi, et al. "EE-LLM: Large-Scale Training and Inference of Early-Exit Large Language Models with 3D Parallelism." Forty-first International Conference on Machine Learning (ICML), 2024.}$
> $\tiny[2] \text{Pan, Xuchen, et al. "EE-Tuning: An Economical yet Scalable Solution for Tuning Early-Exit Large Language Models." CoRR (2024).}$

---

### Meta-Review · Area_Chair_SsJV · 2026-01-09

**Summary:**

The paper proposes hierarchical speculative decoding (HiSpec), which adds an intermediate verification step to early-exit speculative decoding (EESD) frame and demonstrates that it improve the throughput over EESD without intermediate verification. The reviewers liked the novelty and writing of the paper.

The main concern from the reviewers is about the generalizability of the approach, which I concur. The main novelty of the paper is to add an intermediate step to EESD method and system design around it, which limits its usage to EESD scenarios. The initial experiments focused on models with existing trained early-exit functionalities. While the paper added experiments on fine-tuned models, the result is only compared with vanilla AR generation instead of EESD and hence further investigation needs to be conducted. Moreover, the paper would also benefit from comparison to non-EE SD methods, or discuss how the method could be combined with non-EE SD methods, which would greatly improve the paper's scope, especially given that many SOTA results are not achieved by EESD methods [1].

Hence. I would recommend a reject for the paper.

[1] Unlocking Efficiency in Large Language Model Inference: A Comprehensive Survey of Speculative Decoding
Heming Xia, Zhe Yang, Qingxiu Dong, Peiyi Wang, Yongqi Li, Tao Ge, Tianyu Liu, Wenjie Li, Zhifang Sui

**Reviewer Concerns:**

The following concerns are partly addressed by the rebuttal: (1) generality of the method beyond Llama family of models; (2) further details on the experiments including acceptance accuracy, speculation lengths, and etc.; (3) comparison with other methods such as SPRINTER.

The generality of the proposed approach still needs further justification.

**Reviewer Scores:**

I think reviewer ZW4r and reviewer cjaU could have moderate chance of bumping up their score to 6. But I feel that would not change the final recommendation.

---

### Decision · Program_Chairs · 2026-01-26

Reject